# Effect of Estrogen on Sirt1 Signaling in Human Macrophages

**DOI:** 10.3390/ijms26178670

**Published:** 2025-09-05

**Authors:** Maria Luisa Barcena, Anne Breiter, Julia Temp, Yury Ladilov, Vera Regitz-Zagrosek

**Affiliations:** 1Department of Urology, Eberhard Karls University of Tuebingen, 72074 Tuebingen, Germany; 2Berlin Institute of Health, Institute for Gender in Medicine, Charité—Universitätsmedizin Berlin, Corporate Member of Freie Universität Berlin, Humboldt-Universität zu Berlin, 10117 Berlin, Germanyvera.regitz-zagrosek@charite.de (V.R.-Z.); 3Faculty of Medicine, Danube Private University, 3500 Krems an der Donau, Austria; 4Brandenburg Medical School Theodor Fontane, Department of Cardiovascular Surgery, Heart Center Brandenburg, University Hospital, 16321 Bernau bei Berlin, Germany; yury.ladilov@rub.de; 5Department of Cardiology, University Hospital Zürich, University of Zürich, 8091 Zürich, Switzerland

**Keywords:** Sirt1, acetylation, macrophage-like cells, sex differences, nucleus isolation

## Abstract

Cardiovascular diseases are the leading cause of human death worldwide. The role of the female hormone estrogen (E2) in this context is subject of debate. E2 might counteract inflammation by acting on macrophages; however, the underlying cellular mechanisms remain poorly understood. In the current study, we used primary human macrophages to investigate the effects of E2 on the NAD^+^-dependent deacetylase Sirt1, protein acetylation, and pro-inflammatory phenotype. Male and female primary monocytes from healthy adult individuals were polarized into pro-inflammatory M1 macrophages via treatment with LPS and IFN-γ followed by treatment with E2 for 24 h. While E2 treatment had no effect on the Sirt1 protein expression, it significantly increased the acetylation state of nuclear proteins p53 and Ku70. In addition, E2 increased NFκB-p65 expression exclusively in male M1 macrophages, while TNF-α was reduced in female M1 macrophages following E2 treatment. In male monocyte-like cells, E2 significantly reduced nuclear Sirt1 expression and increased Ku70 acetylation. The current study demonstrated that E2 treatment of human M1 macrophages leads to downregulation of nuclear Sirt1 and hyperacetylation of corresponding nuclear proteins. These molecular changes are associated with an enhancement of the pro-inflammatory phenotype in male primary macrophages, while an attenuation of inflammation was observed in female cells.

## 1. Introduction

Cardiovascular diseases are the leading cause of death worldwide, with men being affected more frequently at a younger age, particularly in the case of coronary heart disease [1]. In women, cardiovascular diseases usually occur later, possibly due to the protective effect of the sex hormone estrogen (E2), which may influence inflammatory processes and acts on chemokines and cytokines [2,3,4,5,6]. Immune cells, specifically macrophages, play a pivotal role in healthy and diseased hearts [7,8].

Monocytes can differentiate into pro-inflammatory M1 macrophages via interferon γ (IFN-γ) or lipopolysaccharide (LPS) stimuli, which promotes the release of pro-inflammatory mediators such as tumor necrosis factor α (TNF-α) and interleukin 6 (IL-6) [9,10]. E2 may affect immune cells, i.e., monocytes and macrophages, and influence the inflammatory response in these cells [11,12] and mainly exhibits anti-inflammatory and protective effects [13,14,15]. In particular, in human macrophages, E2 inhibits expression of pro-inflammatory cytokines via inhibition of the nuclear transcription factor ‘kappa-light-chain-enhancer’ of activated B-cells (NFκB) [13,16]. In a previous study, we also demonstrated sex-specific differences in E2’s modulation of the inflammatory profiles during pro-inflammatory M1/ anti-inflammatory M2 polarization in murine bone marrow macrophages [17]. However, underlying molecular mechanisms remain poorly understood.

Sirtuins play a crucial role in regulating macrophage function, particularly under inflammatory environments [18,19]. Activation of the NAD^+^-dependent deacetylase sirtuin 1 (Sirt1) signaling exerts pronounced anti-inflammatory effects in macrophages [20]. E2 may regulate the expression and function of Sirt1. Several studies have demonstrated that estrogen receptor (ER)α directly binds to the Sirt1 promoter thereby upregulating its expression in breast cancer [21,22,23]. Similarly, E2 has been shown to increase the Sirt1 expression in endothelial cells, conferring protective effects both in vitro and in animal models [24]. Activated Sirt1 deacetylates its target proteins, including the transcription factor NFκB, and, therefore, mitigating inflammation [25,26,27]. Moreover, E2 treatment enhances the expression and activity of mitochondrial Sirt3, which alleviates both the oxidative and pro-inflammatory phenotype of pro-inflammatory M1 macrophages [28]. Whether E2 may affect Sirt1 activity and expression in human male and female M1 macrophages remains largely unknown.

Therefore, in the present study, we investigated sex-related E2 effects on the expression and activity of Sirt1 in pro-inflammatory human macrophages. The analyses revealed an increased acetylation of nuclear proteins, which was accompanied by the enhanced pro-inflammatory phenotype in male macrophages, while its attenuation was found in female macrophages.

## 2. Results

### 2.1. Male and Female Human Primary M1 Macrophages Express Estrogen Receptors

To investigate the effect of E2 on the M1 polarization of human primary macrophages derived from monocytes, the expression of the estrogen receptors (ER) in male and female cells was analyzed. The three known ERs, i.e., ERα, ERβ, and GPR30, are expressed in male and female macrophages at the protein level (Appendix A). The E2-dependent activity of the ER can be proved via ERK1/2 activation [29,30]. Our results show a sex-independent increased ERK1/2 phosphorylation in human macrophages after 24 h E2 treatment (Appendix A).

### 2.2. Sex Differences in the E2 Effects on Human Primary Pro-Inflammatory M1 Macrophages

To investigate sex differences in the M1 polarization of human primary macrophages, monocytes from male and female donors were polarized into pro-inflammatory M1 macrophages by treatment with LPS and IFN-γ, and prominent pro-inflammatory markers were analyzed. LPS and IFN-γ treatment significantly increased TNF-α expression at the RNA level in both sexes (Figure 1B). In addition, the IL-1β expression was significantly increased in male and female M1 macrophages, however its expression was much higher in male pro-inflammatory macrophages (Figure 1C). In contrast, CD80 expression was exclusively increased in female macrophages after LPS and IFN-γ treatment (Figure 1D).

In the next step, we investigated the effects of E2 on inflammatory markers in human primary M1 macrophages following LPS and IFN-γ treatment. The E2 treatment significantly increased the NFκB expression only in male primary M1 macrophages, while in female macrophages, E2 significantly decreased the expression of the pro-inflammatory marker TNF-α (Figure 2B,C). The expression levels of IL-1β and CD80 were not affected after E2 treatment in either in male or in female macrophages (Figure 2D,E).

### 2.3. E2 Treatment Increases Acetylation of Sirt1 Targets in Primary Human M1 Macrophages

To determine whether the observed effects on the inflammatory phenotype were associated with the alterations in Sirt1 signaling, we examined Sirt1 expression as well as protein acetylation. E2 treatment did not affect Sirt1 expression and total cellular protein acetylation in either male or female cells (Figure 3A–D). Nevertheless, a more detailed acetylation analysis of the potential Sirt1 targets revealed significant hyperacetylation of p53 in female cells and Ku70 in both male and female cells (Figure 3E–H).

### 2.4. E2 Modulates the Nuclear Sirt1 Expression in Male Human M0 and M1 Monocyte-like Cells

To investigate the action of E2 independently of possible epigenetic effects in primary macrophages, human male monocyte-like THP-1 cells were used. While E2 treatment did not affect the Sirt1expression in the full cell lysates or in the nucleus-free fraction of M0 and M1 THP-1 cells (Figure 4A,B,D), Sirt1 expression was significantly reduced in the nuclear fraction of M0 THP-1 cells after E2 treatment (Figure 4A,C). Analysis of the effects of E2 on the acetylation rate of Ku70 in THP-1 cells revealed increased Ku70 acetylation in the full cell lysates of pro-inflammatory M1 THP-1 cells (Figure 4E,F). In the nuclear fraction, E2 treatment significantly increased Ku70 acetylation in M0 THP-1 cells (Figure 4G,H).

## 3. Discussion

In the present study, we aimed to investigate the effects of E2 on Sirt1 expression and activity in human male and female macrophages and its effect on the macrophage phenotype. Our findings revealed that (i) E2 promoted the inflammatory phenotype in male primary M1 macrophages, while it alleviated inflammation in female cells. (ii) E2 did not affect Sirt1 expression or total protein acetylation in primary M1 macrophages but led to significant hyperacetylation of nuclear Sirt1target proteins p53 and Ku70. (iii) A similar hyperacetylation of Ku70 was observed in male monocyte-like THP-1 cells following E2 treatment.

Sirt1 is a NAD^+^-dependent deacetylase involved in the regulation of numerous cellular processes, including mitochondrial biogenesis, autophagy, inflammatory response, and aging processes [31,32]. The role of sirtuins in determining macrophage phenotype and function has been demonstrated in several reports [33]. Moreover, E2 appears to profoundly modulate immune cells, such as monocytes and macrophages [28,34,35] and has effects on the inflammatory response [36,37]. Although pro-inflammatory actions of E2 have been reported, most of its actions are anti-inflammatory [38,39]. These discrepancies may be attributed to the type of cell or tissue [40,41].

To address the molecular basis of these sex differences, it is essential to consider the distinct roles of ER subtypes and their impact on Sirt1 expression or activity. Previous studies suggest that the Sirt1-upregulating effects of E2 are mediated via ERα [21] and G protein-coupled estrogen receptor (GPER30), especially in female macrophages [42]. In contrast, different expression of ERβ and Sirt1 may partly explain the pro-inflammatory responses observed in male cells [43]. Consequently, the sex-specific impact of E2 on macrophage phenotype and nuclear acetylation likely reflect a dynamic interplay between ERα, ERβ, and GPER30 subtypes, contextually regulated by cell type and environment.

Applying male and female primary human macrophages isolated from healthy individuals of similar age, we found comparable expression of all three estrogen receptors and similar cellular responsiveness to the E2 treatment as evidenced ERK1/2 phosphorylation- a widely used functional test [30,44]. In accordance, several studies have proposed that E2 acts via ER signaling on monocytes and macrophages [17,28,45], suggesting that E2 directly or indirectly influences the macrophage phenotype [28,46,47].

In the present study, the treatment of the primary human macrophages with E2 resulted in hyperacetylation of the Sirt1 target p53, but notably, this effect was observed exclusively in female macrophages. This suggests a sex-specific regulation of p53 acetylation in response to estrogen. Consistently, studies by Solomon et al. demonstrated that the specific Sirt1-inhibitor EX-527 significantly increases p53 acetylation at lysine 382. Interestingly, despite this increase, Sirt1 inhibition did not affect cell growth, viability, or p53 controlled gene expression following etoposide-induced DNA damage [48].

Furthermore, our data show that E2 treatment decreased nuclear Sirt1 expression in male monocyte-like THP-1 cells, which was associated with hyperacetylation of the nuclear Ku70 protein. In line with these findings, Jeong et al. reported that Sirt1 physically interacts with Ku70, promoting its deacetylation and thereby enhancing DNA repair capacity. Overexpression of Sirt1 was shown to increase DNA strand break repair, while its repression reduces this activity, indicating a direct and critical role in DNA repair processes [27].

In the present study, treatment of primary human macrophages with E2 did not alter Sirt1 expression or total protein acetylation. However, more specific analysis of nuclear protein acetylation revealed a significant hyperacetylation of the Sirt1 target p53, which was observed exclusively in female macrophages. Previous studies have demonstrated that Sirt1 inhibition leads to hyperacetylation of p53, disrupting the Sirt1-p53 interaction and contributing to Cr(VI)-induced ovarian apoptosis [49]. Using the THP1 macrophage cell line, we further found that the downregulation of Sirt1 in M0 cells was accompanied by hyperacetylation of nuclear Ku70. Several studies suggest that E2 may differentially regulate Sirt1 expression, depending on tissue and cell type [21,50,51]. For instance, in female vascular smooth muscle cells, E2 treatment results in to Sirt1 downregulation, paired with increased apoptosis and reduced proliferation, mediated via Akt and ERK pathways [52]. Conversely, E2 has been shown to protect the postnatal rat brain against acute ethanol intoxication by activating Sirt1 [53]. Similarly, in mouse bone marrow adipocytes, absence of E2 was associated with reduced Sirt1 levels, whereas E2 replacement restored Sirt1 expression [54]. Thus, the effects of E2 on Sirt1 depend on tissues and cell types. Beyond Sirt1, E2 may also upregulate other sirtuins [28] and HDACs [55], which could further influence the acetylation of nuclear proteins.

Several studies proposed E2-dependent modulation of inflammatory actions in immune cells [34,56]. Accordingly, we found an estrogen-dependent increased in the pro-inflammatory response in male macrophages, whereas E2 appeared to exert anti-inflammatory effects in female macrophages. This aligns with previous reports indicating that E2 promotes pro-inflammatory activity in men while it acts as an anti-inflammatory agent in women [57]. In accordance, E2 treatment produced exclusively pro-inflammatory effects in male murine bone marrow macrophages [17]. Moreover, another study demonstrated that serum TNF-α levels are lower in from premenopausal women compared to men and postmenopausal women [58]. In female monocytes, E2 tends to have anti-inflammatory effects [16,59]. Additionally, research has shown that the expression of antioxidant enzymes in cardiac tissue of premenopausal women is higher than in men, although this sex-related difference diminishes with age [60]. Taken together, these findings suggest that E2 can be detrimental in male macrophages and protective in female cells.

### Limitations of the Study

In this study, we observed an estrogen-dependent increase in NFκB expression in male human monocytes; however, we did not assess NFκB activation, so the results should be interpreted with caution. Nonetheless, we speculate that E2 modulates NFκB activity, as several studies have demonstrated the involvement of E2 in regulating the NFκB signaling pathway [61,62,63].

## 4. Materials and Methods

### 4.1. Isolation and Cultivation of Human Monocyte-Derived Macrophages

Peripheral blood samples (80 mL each) were collected from healthy young adults (14 men, 13 women, aged 19–58 years,) for the isolation of peripheral blood mononuclear cells (PBMCs) (Appendix A). We obtained informed consent from all study participants. Sample collection and the experimental protocols were approved by the Scientific Board of the Charité—Universitätsmedizin Berlin (EA1/070/16). All experiments were conducted in accordance with the German regulations and the ethical standards outlined in the Declaration of Helsinki. PBMCs were isolated using Ficoll (Sigma-Aldrich, Darmstadt, Germany) density centrifugation from freshly collected blood [64]. Monocytes were isolated from PBMCs using human CD14 MicroBeads (Catalog no. 130-050-201, Miltenyi Biotec, Bergisch Gladbach, Germany) and seeded on 6-well plates (1 × 10^6^ cells/well) in phenol-red free RPMI 1640 medium (Catalog no. 11835030, Gibco, Grand Island, NY, USA) supplemented with 4 ng/mL human macrophage colony-stimulating factor (M-CSF) (Catalog no. 300-25-10, PeproTech, Hamburg, Germany), 10% Fetal Bovine Serum (FBS) Advanced (Catalog no. FBS-11A, Capricorn Scientific, Ebsdorfergrund, Germany) and 1% penicillin/streptomycin (Catalog no. A2210, Biochrom, Berlin, Germany) at 37 °C with 95% humidity and 5% CO_2_ for 72 h.

### 4.2. In Vitro Studies with Human Monocytic THP-1 Cells

Human monocytic THP-1 cells (DSMZ No.: ACC16, Germany) were routinely cultured in RPMI 1640 medium (Catalog no. 72400021, Gibco, Grand Island, NY, USA) supplemented with L-glutamine, 10% FBS Advanced (Catalog no. FBS-11A, Capricorn Scientific, Ebsdorfergrund, Germany) and 1% penicillin/streptomycin (Catalog no. A2210, Biochrom, Berlin, Germany) at 37 °C and 5% CO_2_ [28,59]. Exponentially growing cells were harvested by centrifugation (5 min/300 rpm) and resuspended in fresh medium every 2 days [64].

### 4.3. Macrophage Polarization

For pro-inflammatory priming, human primary monocytes or THP-1 cells were treated with LPS (10 ng/mL; Catalog no. L6529, Sigma, Darmstadt, Germany) and recombinant human IFN-γ (10 ng/mL; Catalog no. 300-02-20, PeproTech, Hamburg, Germany) for 24 h [17,64]. Untreated cells were used as controls.

### 4.4. Activation of ERs

For ER activation experiments, LPS/IFN-γ stimulated human primary monocytes or THP-1 cells were starved with a phenol-free medium and 2.5% charcoal-stripped FCS (Catalog no. P30-2302, PAN Biotech, Frankfurt, Germany) for 24 h prior to E2 treatment. After starvation, cells were treated with 10 nmol/L water soluble E2 (Catalog no. E4389-100MG, Sigma-Aldrich, Darmstadt, Germany) or 10 nmol/L dextrin (Catalog no. C0926-5G, Sigma-Aldrich, Darmstadt, Germany) as vehicle control for 24 h [17].

### 4.5. RNA Isolation and Quantitative Real Time (RT)-PCR

Total RNA was isolated using RNA-Bee™ (Catalog no. CS-105B-5, Thermo Fisher, Bremen, Germany). Quantitative real-time PCR was conducted using the Brilliant SYBR Green qPCR master mix (Catalog no. 4368706, Applied Biosystems, Waltham, MA, USA). Relative mRNA levels were determined using the comparative threshold cycle (Ct) method as described previously [65]. The target gene expression was normalized to the expression of hypoxanthine phosphoribosyl transferase (HPRT) or ribosomal protein lateral stalk subunit P0 (RPLP0). All samples were analyzed as biological triplicates and technical duplicates.

### 4.6. Western Blot Analysis

Cells were homogenized in Laemmli-Buffer [66]. Protein concentrations were determined with the Pierce™ 660 nm Protein Assay (Catalog no. 22660; Thermo Scientifc, Bremen, Germany). Equal amounts of total proteins were separated by SDS-PAGE and transferred to a nitrocellulose membrane (Catalog no. 4675.1, Karlsruhe, ROTH, Germany). The membrane was incubated overnight with primary antibodies: ERα (H-184, #sc-7202, 1:200, Santa Cruz, CA, USA), ERβ (1531, #sc-53494, 1:500, Santa Cruz, CA, USA), GRP30 (N-15, #sc-48525, 1:1000, Santa Cruz, CA, USA), ERK (p44/42, #4695S, 1:1000, Cell Signaling, Danvers, MA, USA), pERK (p44/42, #4370S, 1:2000, Cell Signaling, Danvers, MA, USA), Sirt1 (#8469, 1:1000, Cell signaling, Danvers, MA, USA), p53 (#sc-126, 1:200, Santa Cruz, CA, USA), ac-p53 (Lys382, #RK2298798, 1:500, Thermo Fisher, Bremen, Germany), Ku-70 (N3H10, #sc-56129, 1:200, Santa Cruz, CA, USA), ac-Ku-70 (acetyl K331, #ab190626, 1:2000, Abcam, UK), ac-lysine (#9411S, 1:2000, Cell signaling, Danvers, MA, USA), NFκB p65 (F-6, #sc8008, 1:200, Santa Cruz, CA, USA), and TIM-23 (#611222, 1:3000, BD Biosciences, Heidelberg, Germany). Reference proteins used to confirm equal loading in the different compartments were α-Tubulin (#3873, 1:10,000, Cell Signaling, Danvers, MA, USA), GAPDH (#MAB374, 1:50,000, EMD Millipore, Darmstadt, Germany) or Lamin B1 (#13435S, 1:1000, Cell Signaling, Danvers, MA, USA). Immunoreactive proteins were detected using ECL Plus reagent (GE Healthcare, Buckinghamshire, UK) and quantified with ImageLab (version 5.2.1 build 11, Bio-Rad Laboratories, Hercules, CA, USA). Samples were run on different gels. Original Western blot data are provided in the Appendix A.

### 4.7. Isolation of Pure Intact Nuclei

Isolation of pure intact nuclei was performed as previously described by Rosner et al. [67]. Briefly, 20 µL of the 25x protease inhibitor cocktail and 10 µL of the 100 mmol phenylmethylsulfonyl fluoride (PMSF) were added to all buffer solutions.

For each preparation, 5 × 10^6^ THP1 cells were used. Cells were washed with ice-cold PBS, then resuspended in cytoplasmic extraction buffer (20 mmol TRIS (pH 7.6); 0.1 mmol EDTA; 2 mmol MgCl_2_·6H_2_O; 0.5 mmol NaF; 0.5 mmol Na_3_VO_4_). After 12 min incubation, cells were homogenized by addition of 35 μL of 10x IGEPAL CA630 and then centrifugated at 4 °C, 500× *g* for 3 min. Nuclei were collected as pellets, the supernatant represented the cytoplasmic fraction.

The pellet was washed in 1x IGEPAL CA-630, gently mixed and centrifuged at 4 °C and 500× *g* for 3 min. Nuclear pellets were then resuspended in nuclear extraction buffer (20 mmol HEPES (pH 7.9); 400 mmol NaCl; 25% Glycerol (*v*/*v*); 1 mmol EDTA; 0.5 mmol NaF; 0.5 mmol Na_3_VO_4_; 0.5 mmol DTT), snap frozen twice and incubated on ice for 20 min. Soluble nuclear proteins were separated from the insoluble fraction by high-speed centrifugation at 4 °C and 20,000× *g* for 20 min. The purity of nuclear fraction was confirmed by Western blot using nuclear-specific markers (Appendix A).

### 4.8. Flow Cytometry

The purity of isolated CD14^+^ human monocytes was assessed by flow cytometry using human anti-CD45-VioBlue (Catalog no. 130-110-637, Miltonyi Biotec, Bergisch Gladbach, Germany) and human anti-CD14-APC-Vio770 (Catalog no. 130-113-706, Miltonyi Biotec, Bergisch Gladbach, Germany), following to the manufacturer’s protocol [17] (Figure 1A). Acquisition was performed on a MACS-Quant flow cytometer (Miltenyi Biotec, Germany) data analysis was carried out using FlowJo software v10.10.

### 4.9. Statistical Analysis

Data were evaluated using the non-parametric Mann–Whitney test. Statistical significance was defined as *p* < 0.05. Statistical analysis was performed by using Prism 10 for Windows (GraphPad Software 10, San Diego, CA, USA).

## 5. Conclusions

In conclusion, our study demonstrated that E2 treatment of human M1 macrophages results in downregulation of nuclear Sirt1 and subsequent hyperacetylation of nuclear proteins. This molecular change was associated with an enhanced pro-inflammatory phenotype in male primary macrophages, whereas an attenuation of this phenotype was observed in female cells.

## Figures and Tables

**Figure 1 ijms-26-08670-f001:**
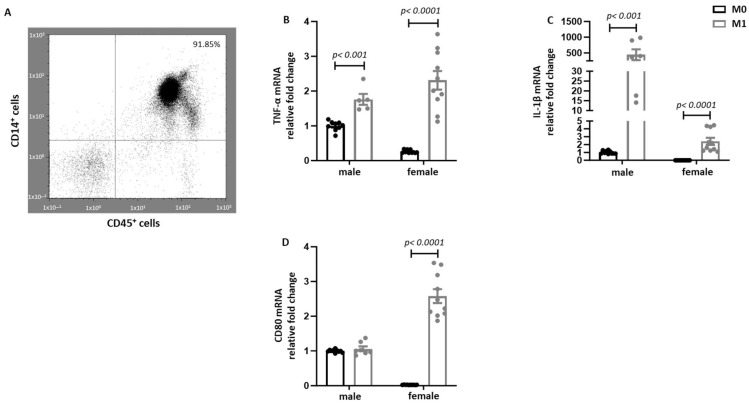
**Characterization of human monocyte-derived macrophages.** (**A**) FACS analysis of the monocyte population purity with CD45-VioBlue and CD14-APC-Vio 770 antibody after isolation of primary human monocytes. Real-time PCR analyses of (**B**) TNF-α, (**C**) IL-1β, and (**D**) CD80 performed with M0 and M1 differentiated human macrophages from male and female individuals. Data are shown as mean ± SEM (n = 5–10). Data were normalized to male M0 macrophages. M0 macrophages: untreated macrophages; M1 macrophages: 10 ng/mL LPS and 10 ng/mL IFN- γ treatment for 24 h.

**Figure 2 ijms-26-08670-f002:**
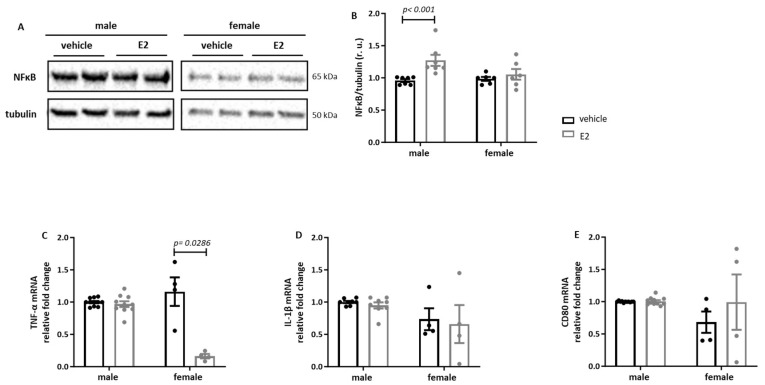
**E2-dependent modulation of the expression of pro-inflammatory markers in human monocyte-derived macrophages.** Western blot analyses and statistics of (**A**,**B**) NFκB and real-time PCR analyses of (**C**) TNF-α, (**D**) IL-1β, and (**E**) CD80 performed with lysates of primary human M1 macrophages treated with E2 (10 nmol/L) for 24 h. Data are shown as the means ± SEM (n = 4–11). All data were expressed in relative units (r.u.). M1 macrophages: 10 ng/mL LPS and 10 ng/mL IFN- γ treatment for 24 h.

**Figure 3 ijms-26-08670-f003:**
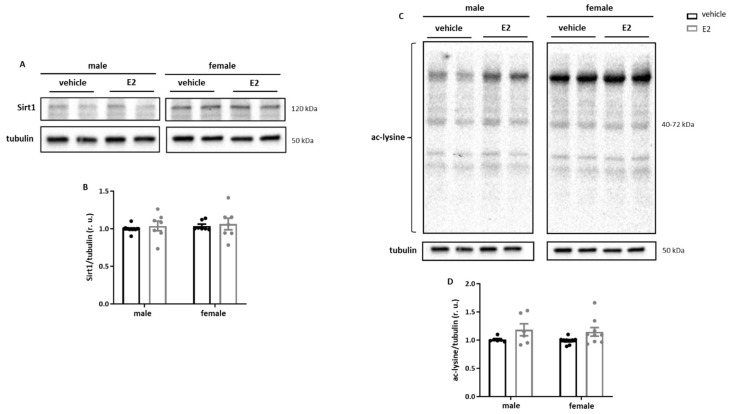
**E2-dependent hyperacetylation of p53 and Ku70 in human monocyte-derived macrophages.** Western blot analysis and statistics of (**A**,**B**) Sirt1, (**C**,**D**) acetylated-lysine, (**E**,**F**) acetylated p53/p53, and (**G**,**H**) acetylated Ku70/Ku70 performed with lysates of primary human M1 stimulated macrophages after 24 h treatment with E2 (10 nmol/L). Data are shown as the means ± SEM (n = 6–13). All data were normalized to the corresponding control and expressed in relative units (r.u.). M1 macrophages: 10 ng/mL LPS and 10 ng/mL IFN- γ treatment for 24 h.

**Figure 4 ijms-26-08670-f004:**
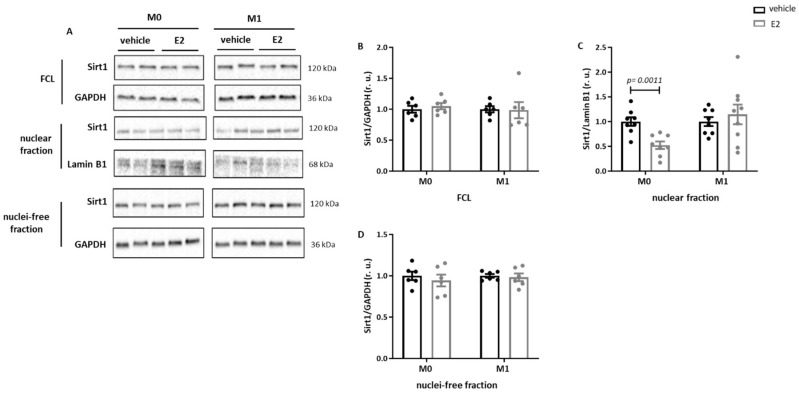
**E2 effects on the Sirt1 expression and Ku70 acetylation in THP-1 cells.** Western blot analysis and statistics of Sirt1 in (**A**,**B**) full cell lysate (FCL), (**A**,**C**) nuclear fraction, (**A**,**D**) nuclei-free fraction and (**E**,**F**) acetylated Ku70/Ku70 in full cell lysate (FCL) and (**G**,**H**) in nuclear fraction of THP-1 cells after 24 h treatment with E2 (10 nmol/L). Data are shown as the means ± SEM (n = 5–8). All data were normalized to the corresponding control and expressed in relative units (r.u.).

## Data Availability

The original contributions presented in the study are included in the article. Further inquiries can be directed to the corresponding author.

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
