# Peer review of "Effect of Estrogen on Sirt1 Signaling in Human Macrophages"

_ijms, 2025, doi:10.3390/ijms26178670_

Round 1

Reviewer 1 Report

Comments and Suggestions for Authors

Major comments:
1. Figure 1A is not clear to observe or read, particularly the axis labels. Please provide a higher-quality image with larger font sizes for the axis labels.

2. While ERK1/2 phosphorylation confirms general ER activity, different ERs can mediate distinct or even opposing effects. A brief discussion on which specific ERs might be mediating these sex-specific inflammatory and Sirt1-related effects could be valuable.

Minor comments:
1. The authors are requested to provide full original images for all the Western blot data in the supplementary.
2. See comments under the language section for suggested minor edits

Comments on the Quality of English Language

The language appears to be of acceptable quality, but can be further improved in certain sections of results and discussion. Consider rephrasing and using short sentences for better reading flow and impact. The following are a few notes to consider:

1. "E2 increased the expression of NFκB-p65 only in male M1 macrophages, while it markedly decreased the pro-inflammatory marker TNF-α in female M1 macrophages."

2. Results section: "it is much stronger express in male pro-inflammatory macrophages" should be edited to "it is much stronger expressed" or "it shows much stronger expression."

3. Introduction section: "accompanied by the enhanced the pro-inflammatory phenotype" should be edited to "accompanied by the enhanced pro-inflammatory phenotype."

4. Discussion section: "Sir1" should be consistently edited to "Sirt1" as used elsewhere in the manuscript.

Author Response

Major comments:

1. Figure 1A is not clear to observe or read, particularly the axis labels. Please provide a higher-quality image with larger font sizes for the axis labels.

Response: Figure 1 has been improved.

2. While ERK1/2 phosphorylation confirms general ER activity, different ERs can mediate distinct or even opposing effects. A brief discussion on which specific ERs might be mediating these sex-specific inflammatory and Sirt1-related effects could be valuable.

Response: We thank the reviewer for the valuable suggestion. We have addressed this issue in the discussion section: “To address the molecular basis of these sex differences, it is essential to consider the distinct roles of ER subtypes and their impact on Sirt1 expression or activity. Previous studies suggest that the Sirt1-upregulating effects of E2 are mediated via ERα [21] and G protein-coupled estrogen receptor (GPER30), especially in female macrophages [42]. In contrast, different expression of ERβ and Sirt1 may partly explain the pro-inflammatory responses observed in male cells [43]. Consequently, the sex-specific impact of E2 on macrophage phenotype and nuclear acetylation likely reflects a dynamic interplay between ERα, ERβ, and GPER30 subtypes, contextually regulated by cell type and environment.”

Minor comments:

1. The authors are requested to provide full original images for all the Western blot data in the supplementary.

Response: We have included the full original images for all Western blot data in the supplementary material.

2. See comments under the language section for suggested minor edits

Response: We thank the reviewer for the advice. We have edited the language section.

Comments on the Quality of English Language

The language appears to be of acceptable quality, but can be further improved in certain sections of results and discussion. Consider rephrasing and using short sentences for better reading flow and impact. The following are a few notes to consider:

1. "E2 increased the expression of NFκB-p65 only in male M1 macrophages, while it markedly decreased the pro-inflammatory marker TNF-α in female M1 macrophages. "In addition, E2 increased NFκB-p65 expression exclusively in male M1 macrophages, while TNF-α was reduced in female M1 macrophages following E2 treatment.”

Response: We have made the correction: “In addition, the IL-1β expression was significantly increased in male and female M1 macrophages, however its expression was much higher in male pro-inflammatory macrophages (Figure 1C)”.

2. Results section: "it is much stronger express in male pro-inflammatory macrophages" should be edited to "it is much stronger expressed" or "it shows much stronger expression."

Response: We have made the correction: “In addition, the IL-1β expression was significantly increased in male and female M1 macrophages, however its expression was much higher in male pro-inflammatory macrophages (Figure 1C)”.

3. Introduction section: "accompanied by the enhanced the pro-inflammatory phenotype" should be edited to "accompanied by the enhanced pro-inflammatory phenotype."

Response: We have made the correction: “Moreover, E2 treatment enhances the expression and activity of mitochondrial Sirt3, which alleviates both the oxidative and pro-inflammatory phenotype of pro-inflammatory M1 macrophages”.

4. Discussion section: "Sir1" should be consistently edited to "Sirt1" as used elsewhere in the manuscript.

Response: We have made the correction. “E2 did not affect Sirt1 expression or total protein acetylation in primary M1 macrophages, but led to significant hyperacetylation of nuclear Sirt1 target proteins p53 and Ku70”.

Reviewer 2 Report

Comments and Suggestions for Authors

The article by Barcena and colleagues, entitled “Effect of Estrogen on Sirt1 Signaling in Human Macrophages”, is well written and presents an interesting study. I do, however, have a few suggestions that may help strengthen the manuscript.

  1. First, the rationale for studying Sirt1 in macrophages could be explained more clearly in the introduction. The authors refer to breast cancer studies, but the direct connection to cardiovascular disease is not well established. Providing a stronger justification for the choice of model system and disease context would improve the overall clarity and impact.
  2. For the Materials and Methods section, I recommend including complete details of all reagents used, catalog numbers, lot numbers, and company names, particularly in the PBMC isolation paragraph.
  3. Regarding Figure 1A, the flow plots should be clear with enlarged labels and more clearly defined gating. In addition, the selection of M0 and M1 macrophage phenotypes requires better explanation in the figure legend, with specific markers indicated.
  4. It would also be helpful to provide clinical demographics for all subjects (both male and female) in a supplementary table. At minimum, please include age, BMI, and any other available clinical characteristics.
  5. For Figure 2, the data on NF-κB signaling are not entirely convincing. The western blot does not clearly show an increase after E2 treatment. To validate signaling activation, the authors could include additional markers, such as phosphorylated p65 and other relevant co-adapter molecules, to confirm NF-κB pathway engagement.
  6. The section titled E2 Treatment Increases Acetylation of Sirt1 Targets in Primary Human M1 Macrophages is interesting, but the western blot figures should include molecular weight markers for clarity.
  7. Please correct the spelling at line 226—it should read Sirt1.

Author Response

Comments and Suggestions for Authors

The article by Barcena and colleagues, entitled “Effect of Estrogen on Sirt1 Signaling in Human Macrophages”, is well written and presents an interesting study. I do, however, have a few suggestions that may help strengthen the manuscript.

1. First, the rationale for studying Sirt1 in macrophages could be explained more clearly in the introduction. The authors refer to breast cancer studies, but the direct connection to cardiovascular disease is not well established. Providing a stronger justification for the choice of model system and disease context would improve the overall clarity and impact.

Response: We thank the reviewer for the valuable suggestion. We have incorporated two relevant studies on cardiovascular diseases to strengthen the rationale in the introduction. “Similarly, E2 has been shown to increase the Sirt1 expression in endothelial cells, conferring protective effects both in vitro and in animal models [24]”.

2. For the Materials and Methods section, I recommend including complete details of all reagents used, catalog numbers, lot numbers, and company names, particularly in the PBMC isolation paragraph.

Response: We have included the catalog numbers of all reagents.

3. Regarding Figure 1A, the flow plots should be clear with enlarged labels and more clearly defined gating. In addition, the selection of M0 and M1 macrophage phenotypes requires better explanation in the figure legend, with specific markers indicated.

Response: Figure 1a and the figure legend have been edited.

4. It would also be helpful to provide clinical demographics for all subjects (both male and female) in a supplementary table. At minimum, please include age, BMI, and any other available clinical characteristics.

Response: The issue raised by the reviewer has been addressed in the supplementary Table 1.

5. For Figure 2, the data on NF-κB signaling are not entirely convincing. The western blot does not clearly show an increase after E2 treatment. To validate signaling activation, the authors could include additional markers, such as phosphorylated p65 and other relevant co-adapter molecules, to confirm NF-κB pathway engagement.

Response: We thank the reviewer for the valuable suggestion. Unfortunately, we are unable to analyze NFκB activity at this time due to insufficient material to perform the necessary experiments. Since we are working with human primary monocytes, recruiting new donors is required, which prevents direct comparison with existing results. Currently, we do not have ethical approval for these additional experiments. Given the importance of this point, we have addressed it in the study’s limitations section. “In this study, we observed an estrogen-dependent increase in NFκB expression in male human monocytes; however, we did not assess NFκB activation, so the results should be interpreted with caution. Nonetheless, we speculate that E2 modulates NFκB activity, as several studies have demonstrated the involvement of E2 in regulating the NFκB signaling pathway (60-62)“.

6. The section titled E2 Treatment Increases Acetylation of Sirt1 Targets in Primary Human M1 Macrophages is interesting, but the western blot figures should include molecular weight markers for clarity.

Response: We have included the molecular weights.

7. Please correct the spelling at line 226—it should read Sirt1.

Response: We have made the correction.